# Racial, Lifestyle, and Healthcare Contributors to Perceived Cancer Risk among Physically Active Adolescent and Young Adult Women Aged 18–39 Years

**DOI:** 10.3390/ijerph20095740

**Published:** 2023-05-08

**Authors:** Jordyn A. Brown, Mahmood A. Alalwan, Sumaya Absie, Naa D. Korley, Claudia F. Parvanta, Cathy M. Meade, Alicia L. Best, Clement K. Gwede, Aldenise P. Ewing

**Affiliations:** 1Department of Epidemiology, Gillings School of Global Public Health, The University of North Carolina, Chapel Hill, NC 27599, USA; jordyn.brown@unc.edu; 2Division of Epidemiology, The Ohio State University College of Public Health, Columbus, OH 43210, USA; alalwan.2@buckeyemail.osu.edu (M.A.A.); absie.2@buckeyemail.osu.edu (S.A.); korley.1@buckeyemail.osu.edu (N.D.K.); 3College of Public Health, University of South Florida, Tampa, FL 33612, USA; cparvanta@usf.edu (C.F.P.); abest@usf.edu (A.L.B.); 4Health Outcomes and Behavior, Moffitt Cancer Center, Tampa, FL 33612, USA; cathy.meade@moffitt.org (C.M.M.); clement.gwede@moffitt.org (C.K.G.)

**Keywords:** perceived cancer risk, adolescent and young adult cancer, women’s health, racial disparities, health promotion

## Abstract

The cancer incidence among adolescents and young adults (AYAs) has significantly increased in recent years, but there is limited information about the factors that influence the perceived cancer risk among AYAs. A cross-sectional, web-based survey of 281 physically active Black and White AYA women was administered to assess the influences of demographic characteristics, family history of cancer, cancer risk factor knowledge, and lifestyle-related risk and protective behaviors on perceived cancer risk. Linear regression analyses were performed in SAS version 9.4. Self-reported Black race (β = −0.62, 95% CI: −1.07, −0.17) and routine doctor visits (β = −0.62, 95% CI: −1.18, −0.07) were related to a lower perceived cancer risk. Family history of cancer (β = 0.56, 95% CI: 0.13, 0.99), cancer risk factor knowledge (β = 0.11, 95% CI: 0.03, 0.19), and current smoking status (β = 0.80, 95% CI: 0.20, 1.40) were related to a higher perceived cancer risk. Perceptions of cancer risk varied among this sample of physically active, AYA women. Lower perceptions of cancer risk among Black AYA women demonstrate a need for culturally tailored cancer educational information that presents objective data on lifetime cancer risk. Reportedly higher perceptions of cancer risk among AYA smokers presents an ideal opportunity to promote smoking cessation interventions. Future interventions to address cancer risk perception profiles among physically active, AYA women should tailor approaches that are inclusive of these unique characteristics.

## 1. Introduction

Cancer remains the second leading cause of death among both men and women in the US, as approximately 1.9 million new cancer cases and approximately 610,000 cancer deaths were projected to occur in 2022 [1]. Most researchers have attributed advanced age as a primary risk factor for the increase in incident cancer cases across populations [2]. However, longitudinal studies now report a marked increase (29.6%) in cancer incidence among adolescent and young adult (AYA) populations ages 15 to 39 [2,3].

In contrast to non-AYA populations, AYA women are more likely to be diagnosed with cancer than AYA men [4,5]. The most common types of cancers diagnosed in AYA women include breast, thyroid, melanoma, cervical, hematological malignancies, and central nervous system tumors [6]. Breast cancer is the most commonly diagnosed malignancy among AYAs in the United States, accounting for nearly 30% of cancers [7].

Approximately 13,000 AYA women are diagnosed with breast cancer annually [8], and AYA women are more likely to die from early-stage breast cancer than older women [6]. Risk factors for AYA breast cancer include family history of cancer, hormonal influences, such as use of contraceptives or older age at first birth, and lifestyle factors, such as smoking and alcohol consumption [7]. Reasons for breast cancer disparities between AYA and non-AYA women are not well understood, but possible explanations include changes in modifiable lifestyle risk factors, in particular obesity, and greater environmental exposures in comparison to previous generations [9].

To improve our understanding of AYA breast cancer across the cancer control continuum, researchers must work to better understand unique health profiles of AYA women. Currently, women at elevated risk include those who engage in risky health behaviors, like limited physical activity, or those with a genetic predisposition. Previous studies have suggested better implementation of multidisciplinary care models to address psychosocial and cancer-related needs across the cancer continuum [8,10]. Notably, interventions aimed at cancer prevention and detection in the pre-diagnosis stage of cancer care and delivery are needed to improve breast cancer-related outcomes in AYA women.

In the present analysis, we ascertained the influences of demographic characteristics, family history of cancer, cancer risk factor knowledge, and engagement in cancer-related risk and protective behaviors on perceived cancer risk among physically active, AYA Black and White women aged 18 to 39 years. Perceived risk is a subjective psychological phenomenon that describes an individual’s susceptibility to disease and probability to benefit from interventions [11,12,13]. The development of interventions based on risk perception profiles may present an opportunity to immediately increase knowledge and awareness of cancer risk, cancer prevention, and early detection—thus reducing the incidence of cancer subtypes, especially breast cancer, among AYA populations [14].

## 2. Methods

### 2.1. Study Design

Following participation in a larger study, physically active AYA women were invited to complete an exploratory, web-based survey to identify cancer-related risk and protective behaviors and personal perceptions about cancer risk given their unique social and health profile between January and April 2019. Additional information regarding the study design has been previously described [14,15]. Briefly, participants were recruited from community partner organizations largely based in the Southeastern US and that hosted recreational sporting events. Recruitment efforts were supplemented by dissemination of recruitment flyers on various social media channels of community partners and a network of universities including Historically Black Colleges and Universities (HBCUs). Informed consent was obtained from all participants included in the study, and participants were given a $10 electronic gift card to a commercial retailer after the questionnaire was submitted to the research team. This study was reviewed by the University of South Florida Institutional Review Board and classified as exempt from federal regulations as outlined by 45 CFR 46.101 (b).

### 2.2. Sample Size and Power Calculations

For this secondary data analysis, we conducted a post-hoc power calculation utilizing specific input parameters since multiple regression analyses were the selected method for this study [16,17]. Power calculation was based on conservative estimates of effect size (f^2^ = 0.15), 0.05 significance level (α = 0.05), an estimated 15 predictors, and an achieved sample size of *n* = 281 participants. Based on these input parameters, it was determined that the 80% power estimate was exceeded.

### 2.3. Study Participants

The inclusion criteria were 18 to 39 years of age, self-reported female sex at birth, self-reported participation in at least one recreational sporting event on average per month, and self-reported racial identity as Black or White. Recreational sports provide an opportunity for young adults to increase their engagement in physical activity alongside fostering social interactions in a seemingly supportive environment. Along with self-reported participation in at least one recreational sporting event per month, women were also asked to report the number of days they engaged in physical activity of at least moderate intensity, which was defined as brisk walking, bicycling at a regular pace, swimming at a regular pace, and heavy gardening in the questionnaire. Together, these criteria were selected to capture a uniquely, physically active population. Furthermore, we only included participants engaging in at least one recreational sporting event per month as AYAs are a highly transient population that are often concerned with short-term disease risk management rather than long-term chronic disease risk management. Future partnership with recreational sporting organizations to promote healthy behaviors and chronic disease prevention may provide the best opportunity to leverage existing healthy behaviors (i.e., physical activity) among this unique subgroup of the population.

Lastly, participants with a previous cancer diagnosis were excluded from the study under the presumption that they would have a higher perception of cancer risk, were abiding by healthcare recommendations, such as consistent screening and follow-up cancer care treatment. Missing values for covariates of interest were removed from the sample. Therefore, we limited our analysis to 281 physically active AYA women who self-identified as White (*n* = 147) or Black (*n* = 134).

### 2.4. Measures

Independent Variables. Independent variables included self-reported demographics (age, education, employment status, income, and marital status), self-reported access to healthcare information (routine doctor’s visit within the past two years, insurance status, and healthcare provider), self-reported family history of cancer, cancer risk factor knowledge score, and self-reported lifestyle behaviors (fruit and vegetable consumption, frequency of physical activity, binge drinking behavior, and current combustible cigarette and e-cigarette smoking status).

Demographics. Age was treated as a continuous variable with normal distribution. The level of education was ordinal and categorized into six groups, namely (1) high school graduate, (2) some college, (3) two-year degree, (4) four-year degree, (5) professional or graduate degree, and (6) doctorate degree. Employment was dichotomized to represent part-time or full-time working status versus otherwise (i.e., unemployed, retired, student, and disabled). The income level was treated as ordinal and classified in increments of $10,000 up to $100,000 followed by the remaining categories: $100,000–$149,999, more than $150,000, and prefer not to answer. The current marital status was dichotomized to represent not being married (i.e., never married, widowed, divorced, or separated) versus otherwise.

Access to Healthcare Information. Three healthcare access-related variables were dichotomized. A routine visit denoted a participant that self-reported attendance at a healthcare visit within the past two years versus otherwise [18]. Self-reported healthcare coverage indicated that the respondent had medical insurance, regardless of public or private designation, versus otherwise. Participant access to a healthcare provider was dichotomized based on self-reported responses to having a medical professional to assess their healthcare needs versus otherwise.

Family History of Cancer. Family history was dichotomized to represent any self-reported first-degree or second-degree family member with a past cancer diagnosis versus otherwise.

Cancer Risk Factor Knowledge. Items on the survey assessing knowledge of cancer-related risk behaviors were adapted from a previous study [19]. The 12 knowledge questions assessed the relationship between certain risk factors, such as smoking, diet, and obesity, and cancer. For example, respondents were asked “Do you think that smoking can increase a person’s chance of developing cancer?” Response options for each knowledge question included “Yes it could”, “No it couldn’t”, and “Don’t know/not sure”. All items were dichotomized to denote the correct response of yes versus otherwise. Correct responses were assigned a value of 1, sum scores were calculated, and an overall average was reported.

Lifestyle Behaviors. Fruit consumption was dichotomized to represent compliance with the recommended daily fruit intake of two cups versus otherwise [20]. Similarly, vegetable consumption was dichotomized to represent compliance with the recommended daily vegetable intake of three cups versus otherwise [20]. Physical activity was categorized into two groups, namely (1) self-reported high engagement characterized by at least four days of physical activity or exercise of at least moderate intensity or (2) self-reported low engagement characterized by three or less days of physical activity or exercise of at least moderate intensity. Binge drinking, hypothesized to be on the causal pathway between recreational sports engagement and sports-related injuries, was categorized into two groups, namely (1) never consuming four or more alcoholic beverages on an occasion in the past 30 days or (2) at least one occasion where a participate consumed four or more alcoholic beverages within the past 30 days [21]. Current combustible cigarette smoking status was dichotomized as either “not risky engagement” characterized by not being a self-reported current combustible cigarette smoker versus “risky engagement” characterized by self-reported combustible cigarette smoking on some or all days. E-cigarette or vaping was categorized into two groups, namely (1) not self-reporting current use of e-cigarette or vaping products or (2) self-reporting current use of e-cigarette or vaping products on some or all days. Lastly, sport participation was dichotomized into individual and team-based sport participation characterized by self-report of belonging to an organized sport activity or league.

### 2.5. Primary Study Outcomes

The perceived cancer risk was the primary outcome of interest for this study. Items comprising a perceived cancer risk scale were adopted from the Health Information National Trends Survey (HINTS) [22]. HINTS items were found to be reliable (a = 0.715) [23] and included the following questions: (1) “How likely are you to get cancer in your lifetime?”; (2) “What would you say is your risk of getting cancer?”; and (3) “Compared to someone else your age and gender, what do you believe your chances are of developing cancer someday?”.

For the first item, response options included a five-point Likert-type scale: “extremely likely” and “somewhat likely” were coded as high (2 points) perceived cancer risk; “neither likely nor unlikely” was coded as moderate (1 point) perceived cancer risk; and “somewhat unlikely” and “extremely unlikely” were coded as low (0 points) perceived cancer risk. Response options were on a four-point Likert-type scale for the second item and included high, moderate, and low/no perceived cancer risk. The last item included response options on a seven-point Likert-type scale where “much higher” and “moderately higher” corresponded to a high perceived cancer risk (2 points); “slightly higher”, “about the same”, and “slightly lower” corresponded to a moderate perceived cancer risk (1 point); and “moderately lower” and “much lower” corresponded to a low perceived cancer risk (0 points).

The perceived cancer risk score was treated as a continuous variable with normal distribution and a higher value representing a higher perceived risk of developing cancer. A sum scale variable on a scale of 0 to 6 points was computed for linear regression analyses.

### 2.6. Statistical Analysis

Descriptive statistics, such as the frequencies and percentages for categorical variables and means (M) and standard deviations (SD) for continuous variables, were computed. Demographic and social characteristics of participants were summarized in total, and differences between the two racial groups were examined using *t* tests and ANOVA for categorical variables and pairwise correlation coefficients for continuous variables. Using linear regression, we estimated the associations between modifiable and non-modifiable independent predictors of perceived cancer risk score. Our model was adjusted for the following a priori selected covariates: age, race, education, employment, family history of cancer, routine doctor visit, healthcare coverage, healthcare provider, cancer risk factor knowledge, fruit and vegetable consumption, physical activity engagement, and current combustible cigarette and e-cigarette smoking status [24,25,26,27,28,29]. Unstandardized beta (β) coefficient with 95% confidence intervals (CIs), standard errors, and t scores were computed to assess the level of association and statistical significance in the multivariate analysis. All statistical analyses were performed in SAS version 9.4 (SAS Inc., Cary, NC, USA).

## 3. Results

The characteristics of the study sample and their respective perceived cancer risk scores are provided in Table 1. The average age of participants was 28.3 years. Racially, there were almost equal proportions of self-reported Black (*n* = 134, 47.7%) and White (*n* = 147, 52.3%) women. Most women were employed (59.4%) and not currently married (73.3%). The largest proportion of the sample received at least a 4-year degree and had an income level less than $50,000. Half of the sample (50.2%) did not have a family history of cancer, and the average cancer risk factor knowledge score was 7.9 out of a possible 12 points. In terms of access to healthcare services, most women self-reported having a routine checkup within the past two years (84.0%), health insurance (91.3%), and sought care from a healthcare provider (87.7%). Most participants were self-reported binge drinkers (81.1%) but did not engage in combustible cigarette (81.9%) or e-cigarette (86.8%) use. Most of the sample did not meet the daily fruit (83.2%) and vegetable (90.8%) intake recommendations. More than half of participants self-reported engagement in physical activity for at least four days of the week (52.5%); nearly 60% of participants reported engagement in individual sport activities, such as running, walking, and cycling.

Significant differences in perceived risk scores were observed for self-reported race, cancer risk factor knowledge, family history of cancer, routine doctor visit, and current combustible cigarette smoking status (Table 1). Black women (M = 2.41, SD = 1.73) reported a lower perceived cancer risk score compared to White women (M = 2.99, SD = 1.83; *p* < 0.05). Participants without a family history of cancer (M = 2.49, SD = 1.90) also reported a lower perceived cancer risk score compared to participants with a family history of cancer (M = 2.94, SD = 1.68; *p* < 0.05). Participants who did not routinely attend a doctor’s appointment (M = 3.22, SD = 1.89) had a higher perceived cancer risk score compared to participants who did attend a doctor’s appointment within the past two years (M = 2.62, SD = 1.77; *p* < 0.05). Current combustible cigarette smokers (M = 3.43, SD = 1.95) also had a higher perceived cancer risk score compared to current non-combustible cigarette smokers (M = 2.56, SD = 1.73; *p* < 0.05).

Self-reported race, family history of cancer, routine doctor visit, cancer risk factor knowledge, and current combustible cigarette smoking status were associated with perceived cancer risk score differences. The β-values presented in Table 2 represent the differences in perceived cancer risk scores. Self-reported Black race (β = −0.62, 95% CI: −1.07, −0.17) and routine doctor visits (β = −0.62, 95% CI: −1.18, −0.07) were related to a lower perceived cancer risk score. A one-point increase in cancer risk factor knowledge was positively associated with a higher perceived cancer score (β = 0.11, 95% CI: 0.03, 0.19). Family history of cancer (β = 0.56, 95% CI: 0.13, 0.99) and current smoking status (β = 0.80, 95% CI: 0.20, 1.40) were related to a higher perceived cancer risk score.

## 4. Discussion

Our study corroborates known differences in perceived cancer risk by self-reported race, family history of cancer, routine doctor visit status, cancer risk factor knowledge, and cigarette smoking status, but extends this knowledge to a new focus on an understudied AYA population of physically active women aged 18–39 years [28,30,31,32,33,34].

Breast cancer is the most diagnosed cancer subtype among AYA populations, and greater consideration for cancer education and behavioral interventions that address both modifiable and non-modifiable risk factors within unique subgroups is needed [35]. Previous research reports that Black women have lower incidence rates of breast cancer in comparison to White women (127.8 versus 133.7 cases per 100,000) [36,37]. Despite lower breast cancer incidence rates, Black women are 40% more likely to die from breast cancer than their White counterparts [36,37]. Additionally, Black women are 41% more likely to be diagnosed with cervical cancer and 75% more likely to die from cervical cancer when compared with White women [38,39]. Racial disparities in breast and cervical cancers also disproportionately affect Black AYA women [8,35,39]. Additionally, our previous research study found our population of physically active women have lower cervical cancer screening rates when compared to national averages of women the same ages [15]. 

Our findings demonstrate the importance of identifying the role of modifiable risk factors (e.g., access to routine doctor visits and cancer risk factor knowledge) and non-modifiable factors (e.g., race and family history). In our sample, more than 80% of participants self-reported having a routine checkup within the past two years, health insurance, and access to a healthcare provider despite a healthcare transition, which is characterized by changes in the provision of healthcare services starting at age 18 years [40]. Healthcare utilization was slightly higher in our sample in comparison to a non-disabled AYA population where only 76% had a preventive visit within the past year and a larger proportion were uninsured (16%) [40]. Similarly, in a community AYA cohort who were between the ages of 15 and 39 years, a little over a third (36%) of the participants reported seeing their primary healthcare provider within the past six months [41]. However, it is important to note that the timing of primary care provider visits occurred in a shorter timeframe when compared to the two-year timeframe of healthcare utilization in our survey.

Both modifiable and non-modifiable risk factors may influence perceived cancer risk, and it is critical to heighten perceived cancer risk of breast and other trending cancers among AYA women. Understanding the contributors for physically active Black women reporting lower perceptions of cancer risk may provide a better understanding of AYAs overall and site-specific cancer trends among Black women. Reasons for lower perceived cancer risk among Black women are multifactorial and related to the persistence of health disparities, in terms of excess mortality relative to other racial and ethnic groups—notably non-Hispanic White women—across multiple chronic health outcomes related to cardiovascular, cancer, and maternal health [42]. As noted by Chinn et al., adverse health outcomes are not independent of the social conditions in which they exist, and this is reflective of the presence of structural inequities that consistently disadvantage Black women from seeking healthcare services and improving their healthcare knowledge [42,43,44]. As a result, many Black women are reluctant to seek out healthcare services and often do not have an opportunity to speak with their primary care provider about cancer risk factors and early signs and symptoms of a cancer diagnosis until an advanced stage cancer diagnosis is made. Earlier and consistent engagement in clinical and community-oriented cancer prevention and cancer detection interventions can further educate and encourage physically active AYA women to pursue both primary and secondary prevention activities that ensure they remain in good health.

Vulnerable subgroup populations of women who smoke, self-identify as Black, are physically active, and between the ages of 18 and 39 years should be included in culturally relevant educational interventions developed to address the unique healthcare needs of AYA populations to reinforce positive health behaviors and, in turn, reduce rising cancer incidence rates. For example, pamphlets, electronic or print handouts, or presentations that provide an overview of cancer risk factors at individual and team recreational sport game locations can provide education in a more relaxed, non-clinical setting. Dissemination of cancer risk factor knowledge at frequently visited exercise locations may be worthwhile to target at times hard-to-reach populations that consider themselves to be in good health.

Another segment of the population that may bypass traditional clinic visits include individuals without a family history of cancer. Previous studies found that participants with a family history of cancer reported a significantly higher perceived cancer risk in comparison to participants with no known family history of cancer [45,46]. This aligns with our findings as half of our sample self-reported a family history of cancer that corresponded to a higher perceived cancer risk score. Family history is also hypothesized to influence perceived cancer risk by age group in some studies as individuals that witnessed their family member’s cancer diagnosis at a younger age have a greater perceived risk of developing cancer [11,47]. Similarly, Lerman et al. found that women between the ages of 30 and 34 had a higher perceived cancer risk compared to other age groups within a population of women with a family history of cancer [48].

A known family history of cancer, heightened perceptions of personal cancer risk, and routine doctor visits can be ideal for promoting cancer screening or early detection interventions [49]. Traditional methods for cancer education are useful as evidenced by high proportions of our sample engaging with healthcare providers and the healthcare system. However, it is important to note that we were able to capture the perspectives of physically active AYA women who participate in physical activity in team settings. This further emphasizes the importance of developing and implementing cancer education activities outside of a clinical setting. For example, we were able to recruit many physically active adults through attendance at in-person sporting events as well as word of mouth across multiple in-person network and virtual social media channels. A recent scoping review by Watson et al. emphasized the importance of building strong interpersonal relationships and appropriate incentives for AYA populations when making healthcare decisions [50]. Collaboration between sports leagues, cancer researchers, and community organizations would be instrumental in the dissemination of cancer risk education as there would be opportunities for relevant stakeholders to share their expertise or form an advisory committee [51,52,53]. Another potential way to engage AYA populations in culturally tailored interventions is through digital interventions, such as webinars and decision sliders, that provide opportunities to supplement face-to-face interactions through the formation of support groups where participants can record their observations and engage in program activities regardless of geographic location [54,55,56]. It is important to balance the value of reaching patients in non-clinical settings, but it is also important to emphasize the influence of routine doctor visits in clinical settings for promoting teachable moments with physically active AYA women who may be at an elevated risk for cancer [57,58].

The association of family history of cancer and perceived cancer risk may also demonstrate the need to develop and implement interventions that address awareness about genetic testing interventions. Pathogenic germline mutations present in up to 17% of breast cancer patients younger than age 45 [8]. Despite improved rates of genetic testing among AYA women, who often have higher rates of regional and distant breast cancer at diagnosis, the rates of cascade testing for family members of genetic mutation carriers, including siblings, are suboptimal [8]. Engaging AYA women in a meaningful way at frequently visited locations, such as the site of their individual or team-based sport activities, in an intervention that encourages consistent cancer screenings or education concerning cancer risk factors can ensure that women at highest risk are reached prior to the late-stage cancer diagnosis.

Our sample of physically active women presents as a uniquely, at-risk group for cancer given that 18% of our sample self-reported being a current smoker in contrast to the 13% of current smokers in the general population [59]. Current research suggests women who begin smoking in adolescence are at a significantly increased risk for breast cancer [60] and 48.5% of deaths from 12 cancer subtypes are associated with cigarette smoking [61,62]. The implementation of intervention activities to reduce the prevalence of cigarette smoking is needed to reduce AYA women’s risk for cancer and other adverse smoking-related health outcomes. Among smokers in our sample, there was a higher perceived cancer risk when compared to non-smokers. This could be attributed to smokers overestimating their cancer risk; however, there are conflicting reports about smokers’ cancer risk perceptions due to measurement error associated with questionnaire design [63]. Although smokers in our sample reported a higher perceived cancer risk, more research is needed to better understand readiness to quit smoking and promote smoking cessation strategies.

Overall, our study assessed cancer risk perceptions within an understudied AYA population of physically active Black and White women between 18 and 39 years of age. Participants were recruited outside of the traditional clinical setting, which presents an alternative pathway to reach AYA population subgroups. Nevertheless, several limitations should be considered. First, our predictors of interest were self-reported, which may be unreliable for sensitive topics, such as family history of cancer and current smoking status, due to limited knowledge of family health or health behavior stigma. Second, we were not able to collect information about other relevant female-dominated cancers, such as breast and cervical cancer, risk factors, such as obesity (measured using BMI), reproductive health outcomes (measured using age at menarche and parity), and premenopausal hormone therapy use due to the scope of the survey instrument. We also did not screen our sample to include the cancer subtypes diagnosed within their family, which can also have implications about the influence and prevalence of hereditary-based cancers among AYA populations.

## 5. Conclusions

Due to rising trends in cancer incidence, more research is needed to better understand the unique social and health-risk related profiles of AYA populations. Efforts to reduce the lifetime risk of cancer among physically active, AYA women aged 18 to 39 may consider cultural (i.e., racially/ethnically appropriate), personalized (i.e., preferred learning/information dissemination style and family history centered conversations with healthcare providers), and lifestyle-focused (i.e., smoking cessation) interventions. Given the discordance in the perceived and actual cancer risk among Black women, cancer public health strategies may consider strengthening community outreach activities and partnerships to provide opportunities to better support the development of cancer risk factor knowledge among the AYA community. We expect our study to help AYA women, clinicians, and researchers to better understand the contribution of modifiable and non-modifiable factors to improve knowledge about cancer risks and outcomes in an at times hard-to-reach population. Further studies to pinpoint the influence of additional demographic, social, and medical characteristics are needed for understanding the current perceived cancer risk among AYA populations.

## Figures and Tables

**Table 1 ijerph-20-05740-t001:** Demographic characteristics of respondents (*n* = 281) and perceived cancer risk score.

	N (col %)	Perceived Cancer Risk Score (Score 0–6)		*p*
		Mean	Standard Deviation	
Overall (*n* = 281)		2.72	1.80	
Age				0.222
Mean = 28.3, SD = 6.2				
Race				0.007 *
White	147 (52.3)	2.99	1.83	
Black	134 (47.7)	2.41	1.73	
Education				0.874
High School Graduate	43 (15.5)	3.00	1.90	
Some College	68 (24.6)	2.72	1.96	
2-year degree	28 (10.1)	2.82	1.36	
4-year degree	87 (31.4)	2.60	1.82	
Professional or Graduate degree	44 (15.9)	2.57	1.66	
Doctorate degree	7 (2.5)	2.71	2.06	
Missing	4			
Employment				0.129
No	114 (40.6)	2.52	1.84	
Yes	167 (59.4)	2.85	1.77	
Income				0.209
Less than $10,000	38 (13.7)	2.53	1.86	
$10,000–$19,999	20 (7.2)	3.20	1.88	
$20,000–$29,999	31 (11.2)	2.29	1.64	
$30,000–$39,999	35 (12.6)	2.14	1.73	
$40,000–$49,999	45 (16.3)	3.29	1.75	
$50,000–$59,999	25 (9.0)	3.20	1.80	
$60,000–$69,999	24 (8.7)	2.71	1.85	
$70,000–$79,999	14 (5.1)	3.00	1.92	
$80,000–$89,999	7 (2.5)	2.86	2.34	
$90,000–$99,999	6 (2.2)	2.67	1.75	
$100,000–$149,999	19 (6.9)	2.26	1.73	
More than $150,000	4 (1.4)	2.25	2.06	
Prefer not to answer	9 (3.3)	2.44	1.24	
Missing	4			
Current Marital Status				0.918
Not Married	203 (73.3)	2.70	1.79	
Married	74 (26.7)	2.73	1.85	
Missing	4			
Cancer Risk Factor Knowledge				0.019 *
Mean = 7.9, SD = 2.8				
Family History of Cancer				0.035 *
No	141 (50.2)	2.49	1.90	
Yes	140 (49.8)	2.94	1.68	
Healthcare Access				
Routine Doctor Visit				0.039 *
No	45 (16.0)	3.22	1.89	
Yes	236 (84.0)	2.62	1.77	
Healthcare Coverage				0.403
No	24 (8.7)	2.42	1.95	
Yes	253 (91.3)	2.74	1.79	
Missing	4			
Healthcare Provider				0.854
No	34 (12.3)	2.76	1.97	
Yes	243 (87.7)	2.70	1.78	
Missing	4			
Lifestyle Factors				
Fruit Consumption				0.235
<2 cups	233 (83.2)	2.77	1.82	
≥2 cups	47 (16.8)	2.43	1.73	
Missing	1			
Vegetable Consumption				0.274
<3 cups	255 (90.8)	2.75	1.80	
≥3 cups	26 (9.3)	2.35	1.81	
Physical Activity				0.208
High Engagement	147 (52.5)	2.58	1.79	
Low Engagement	133 (47.5)	2.85	1.80	
Missing	1			
Binge Drinking				0.452
No	53 (18.9)	2.55	1.67	
Yes	228 (81.1)	2.75	1.83	
Current Smoker				0.002 *
No	230 (81.9)	2.56	1.73	
Yes	51 (18.2)	3.43	1.95	
Current E-cigarette Smoker				0.530
No	243 (86.8)	2.69	1.79	
Yes	37 (13.2)	2.89	1.94	
Missing	1			
Sport Participation Type				0.315
Individual	169 (60.1)	2.63	1.79	
Team-based	112 (39.9)	2.85	1.83	

** p* value for chi-square test for categorical variables. Two-sample *t*-test was used for continuous variables.

**Table 2 ijerph-20-05740-t002:** Multivariable Linear Regression Model Summary.

Model			Unstandardized Coefficients				95% Confidence Interval for β	
			β	Std. Error	t	*p*	Lower Bound	Upper Bound
1	Constant		1.793	0.703	2.55	0.011	0.409	3.177
	Age		0.028	0.020	1.40	0.161	−0.011	0.067
	Race	White	Ref					
		Black	−0.616	0.228	−2.70	0.008 *	−1.065	−0.166
	Education		−0.129	0.087	−1.49	0.138	−0.299	0.042
	Employment	Not employed	Ref					
		Employed	0.288	0.241	1.20	0.232	−0.186	0.762
	Income		−0.034	0.036	−0.94	0.348	−0.106	0.037
	Family History of Cancer	No	Ref					
		Yes	0.561	0.219	2.56	0.011 *	0.129	0.993
	Routine Doctor Visit	No	Ref					
		Yes	−0.624	0.284	−2.20	0.029 *	−1.183	−0.065
	Healthcare Coverage	No	Ref					
		Yes	0.489	0.505	0.97	0.334	−0.506	1.484
	Healthcare Provider	No	Ref					
		Yes	−0.344	0.433	−0.80	0.427	−1.197	0.508
	Cancer Risk Factor Knowledge		0.109	0.040	2.75	0.007 *	0.031	0.186
	Fruit Consumption (≥2 cups)	No	Ref					
		Yes	−0.084	0.296	−0.28	0.777	−0.668	0.499
	Vegetable Consumption (≥3 cups)	No	Ref					
		Yes	−0.397	0.378	−1.05	0.295	−1.140	0.347
	Physical Activity	High engagement	Ref					
		Low engagement	0.229	0.226	1.02	0.310	−0.215	0.673
	Current Cigarette Smoker	No	Ref					
		Yes	0.800	0.304	2.63	0.009 *	0.202	1.398
	Current E-Cigarette Smoker	No	Ref					
		Yes	−0.002	0.323	−0.01	0.994	−0.639	0.634

*** Significant at the *p* < 0.05 level.

## Data Availability

The data presented in this study are available on request from the corresponding author. The data are not publicly available due to privacy and ethical concerns.

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
