# Peer review of "Racial, Lifestyle, and Healthcare Contributors to Perceived Cancer Risk among Physically Active Adolescent and Young Adult Women Aged 18–39 Years"

_ijerph, 2023, doi:10.3390/ijerph20095740_

Round 1
Reviewer 1 Report
I congratulate the authors on what is an interesting manuscript. Only a couple of minor comments that should be addressed:
Methods (Line 74) The authors state “Additional information regarding study design has been 74
previously described”, clearly in two papers already published from this study, However, information on how the participants were sampled would be informative to include in the current manuscript.
Line 91: what were the parameters for moderate physical activity (given these can differ based on setting.
Line 104: why just binge drinking behaviour and not regular alcohol consumption?
Author Response
We thank reviewer 1 for highlighting areas to strengthen within our manuscript. We have addressed each of these comments within our revision. Briefly:
Reviewer Comment: Methods (Line 74) - The authors state “Additional information regarding study design has been previously described,” clearly in two papers already published from this study. However, information on how the participants were sampled would be informative to include in the current manuscript.
Response: Additional information about study recruitment, sampling, and design are included within the methods section (lines 76-80).
Reviewer Comment: (Line 91) - What were the parameters for moderate physical activity (given these can differ based on setting).
Response: Examples of moderate physical activity used in the questionnaire are included within the study participants section (lines 100-102).
Reviewer Comment: (Line 104) – Why just binge drinking behavior and not regular alcohol consumption?
Response: We elected to include binge drinking behavior rather than a categorical alcohol consumption measure (e.g., never/current/former) in our model as the risk of excessive behaviors is worse among recreational sports athletes. In fact, binge drinking has the potential to be on the causal pathway between exposure to a recreational sporting event and sports-related injuries. To emphasize this point, we included an additional sentence to justify the inclusion of binge drinking rather than alcohol consumption in our model (lines 159-163).
Reviewer 2 Report
Brown et al. conducted a cross-sectional study to examine factors that influence perceived cancer risk among AYAs. The main strength of this study includes its novelty, given the emphasis on the AYA population, a highly at-risk population that is disproportionately burdened and for which research is currently lacking. The authors incorporate sociodemographic factors as well as family history of cancer, cancer risk factor knowledge, and engagement in cancer-related risk behaviors. Overall, this is a novel and timely study; please consider the following comments/suggestions to take into account:
Can the authors please specify/include why the study population of “physically active” AYAs were chosen? i.e., Why the inclusion criteria of “at least one recreational sporting event on average per month?”
The authors might reconsider rewording the title. Currently, it is confusing to decipher what exactly is meant by “Physically active, Black AYA”. Given the inclusion criteria for the study population is those who are physically active and physical activity is also a variable analyzed, it is confusing what is meant by this title. Also given the totality of variables related to cancer risk perception, the authors might reconsider only emphasizing one aspect. The authors might want to reconsider wording of regression-based analyses in title.
Can the authors please verify/clarify for the cancer risk factor knowledge questions, “No it couldn’t” and “Don’t’ know/not sure” were combined into one category? The authors then say in the limitations section that “Don’t’ know” were excluded. Please clarify.
In the discussion section, can the authors please provide some more context in terms of the findings “In terms of access to healthcare services, most women self-reported 196 having a routine checkup within the past two years (84.0%), health insurance (91.3%), and 197 sought care from a healthcare provider (87.7%).” More specifically, can the authors discuss/find data comparing these percentages to other studies who have looked at similar outcomes. For context, it would be important to know if this population is similar or distinct to others or is typical in terms of healthcare utilization which may therefore lead to inherent differences in terms of cancer risk perception. There is little mention of the study population and if this is generalizable.
In the discussion section, it would be helpful to include more discussion and possible hypotheses/reasonings behind why certain populations (Black women, those with higher physical activity, etc.) have lower/higher perceived cancer risk. Currently there is a heavy focus on family history of cancer but limited consideration/emphasis on others (which are also emphasized in the title).
Author Response
We thank reviewer 2 for highlighting areas to strengthen within our manuscript. We have addressed each of these comments within our revision. Briefly:
Reviewer Comment: Can the authors please specify/include why the study population of “physically active” AYAs were chosen? i.e., Why the inclusion criteria of at least one recreational sporting event on average per month?
Response: We have included a justification for the inclusion of physically active AYAs. Briefly, we only included participants who self-reported on average at least one recreational sporting event per month due to AYAs being a highly transient population that have fluctuating concerns about short-term and long-term chronic disease risk. Future partnership with recreational sporting organizations may best provide opportunities to reach this subgroup. Lastly, by also assessing days of physical activity, we sought to ensure a uniquely, physically active subgroup of the population to further leverage an already existing healthy behavior for future interventions (lines 96-108).
Reviewer Comment: The authors might reconsider rewording the title. Currently, it is confusing to decipher what is meant by “Physically active, Black AYA.” Given the inclusion criteria for the study population is who are physically active and physical activity is also a variable analyzed, it is confusing what is meant by this title. Also given the totality of variables related to cancer risk perception, the authors might reconsider only emphasizing one aspect. The authors might want to consider wording of regression-based analyses in title.
Response: To better reflect the aims of our study, our manuscript is now entitled “Racial, Lifestyle, and Healthcare Contributors to Perceived Cancer Risk Among Physically Active Adolescent and Young Adult Women Ages 18-39.” We believe it is important to include the “physically active” as a population descriptor to emphasize that we analyzed a unique subset of a potentially underserved population at risk for unfavorable cancer outcomes. In addition, we believe it is important to consider the compound effect of multiple factors, inclusive of modifiable and non-modifiable contributing factors, on perceived cancer risk.
Reviewer Comment: Can the authors please verify/clarify for the cancer risk factor knowledge questions, “No it couldn’t” and “Don’t know/not sure” were combined into one category? The authors then say in the limitations that “Don’t know” were excluded. Please clarify.
Response: This statement was edited within the limitations section to accurately reflect that participants who answered “No it couldn’t” and “Don’t know/not sure” were combined into one category and were given 0 points for their respective responses. Therefore, the participants who responded with “Don’t know/not sure” were included.
Reviewer Comment: In the discussion section, can the authors please provide some more context in terms of the findings “In terms of access to healthcare services, most women self-reported 196 having a routine checkup within the past two years (84.0%), health insurance (91.3%), and 197 sought care from a healthcare provider (87.7%). More specifically, can the authors discuss/find data comparing these percentages to other studies who have looked at similar outcomes. For context, it would be important to know if this population is similar or distinct to others or is typical in terms of healthcare utilization which may therefore lead to inherent differences in terms of cancer risk perception. There is little mention of the study population and if this is generalizable.
Response: This is important to note and as a result, we included additional information about healthcare utilization rates within two populations: (1) a non-disabled AYA population group ages 18-30 and (2) a community cohort of AYAs aged 15-39 to note how our sample compares to the general AYA population.
Reviewer Comment: In the discussion section, it would be helpful to include more discussion and possible hypotheses/reasons behind why certain populations (Black women, those with high physical activity, etc.) have lower/higher perceived cancer risk. Currently, there is a heavy focus on family history of cancer but limited consideration/emphasis on others (which are also emphasized in the title).
Response: This is important to note and as a result, we included additional reasons related to structural health inequities within and outside of the health system that Black women experience throughout the life course. In addition, we included information about a study that hypothesized that there may be an overestimated higher perceived cancer risk among smokers due to measurement error.
Reviewer 3 Report
The manuscript is clearly written with details given for analysis section.
1. There is a need to strengthen the discussion about the findings that Blacks have lower perception of cancer risk, i.e. the contributory factors, especially because this is the title of the article and the main finding of the study
2. There is also a need to increase the discussion on cancer risk perception and age groups, in the context of AYA
3. Recommendations: while the recommendation that culturally tailored cancer educational information is needed, it would complete the thought if the authors can give examples of such information or interventions
Author Response
We thank reviewer 3 for highlighting areas to strengthen within our manuscript. We have addressed each of these comments within our revision. Briefly:
Reviewer Comment: There is a need to strengthen the discussion about the findings that Blacks have lower perception of cancer risk, i.e., the contributory factors, especially because this is the title of the article and the main finding of the study.
Response: This is important to note and was addressed in the previous reviewer comment and response as we included additional information about the generalizability of our results. We included additional reasons related to structural health inequities within and outside of the health system that Black women experience throughout the life course. In addition, we included information about a study that hypothesized that there may be an overestimated higher perceived cancer risk among smokers due to measurement error. To better reflect the aims of our study, our manuscript is now entitled “Racial, Lifestyle, and Healthcare Contributors to Perceived Cancer Risk Among Physically Active Adolescent and Young Adult Women Ages 18-39.”
Reviewer Comment: There is also a need to increase the discussion on cancer risk perception and age groups, in the context of AYA.
Response: This is important to note and was addressed alongside our discussion of family history influence cancer risk perception. This is because participants’ age at a family member’s cancer diagnosis can also have significant implications for their perceived cancer risk.
Reviewer Comment: While the recommendation that culturally tailored cancer educational information is needed, it would complete the thought if the authors can give examples of such information or interventions.
Response: This is important to note, and we included examples of program activities, potential collaborations between stakeholders, and interventions that support our call for culturally tailored cancer education.
Round 2
Reviewer 2 Report
The authors have thoughtfully answered all remaining comments/feedback. Thank you for the opportunity to review this work.
Author Response
We appreciate the opportunity to revise our manuscript “Racial, Lifestyle, and Healthcare Contributors to Perceived Cancer Risk among Physically Active Adolescent and Young Adult Women Ages 18-39.” We have addressed all comments offered by the reviewer to improve the clarity of our manuscript below:
Reviewer Comment: On page 8, be clearer that the b coefficient for self-reported race represents self-reported Black race. Thus, the sentence “Self-reported race (B = -0.62…” should include “Self-reported Black race… were related to lower perceived cancer risk score.”
Response: We have made this change to accurately reflect that the beta coefficient corresponds to self-reported Black race.
Review Comment: Within the table 1 footnotes, please provide information regarding the test statistics used to derive the given p values.
Response: We have included information about how the given p values were derived based on differences in perceived cancer risk score by each respective covariate value.
Reviewer Comment: Within table 2, the term “Sig.” should be changed to p value.
Response: We have made this change in Table 2 so P (representing p-value) is included as a table heading rather than “Sig.”
Reviewer Comment: Given that Black women have increased risks for late-stage breast cancer and likelihood of dying from cancer compared to their non-Hispanic White women counterparts, could the investigators add a sentence within the conclusions (section 5 on pages 10-11) on the public health implications that adolescents and young adult Black women have such a lower perceived cancer risk? For example, “Cancer public health strategies may consider increasing programming and community outreach and engagement partnering with communities of AYA Black women given the discordance in cancer perceived risks and actual risks.”
Response: This is important to note and was addressed in our conclusions section to emphasize higher risk for Black women to be diagnosed with advanced-stage breast cancer and mortality.
Thank you for your time and consideration to carefully review this manuscript for publication within the International Journal of Environmental Research and Public Health.